# A Novel Data Augmentation Method for Improving the Accuracy of Insulator Health Diagnosis

**DOI:** 10.3390/s22218187

**Published:** 2022-10-26

**Authors:** Zhifeng Li, Yaqin Song, Runchen Li, Sen Gu, Xuze Fan

**Affiliations:** State Key Laboratory for Strength and Vibration of Mechanical Structures, School of Aerospace Engineering, Xi’an Jiaotong University, Xi’an 710049, China

**Keywords:** insulator, defect detection, data augmentation, DTW barycenter averaging, adaptive weighting DBA, support vector machines with genetic algorithm

## Abstract

Performing ultrasonic nondestructive testing experiments on insulators and then using machine learning algorithms to classify and identify the signals is an important way to achieve an intelligent diagnosis of insulators. However, in most cases, we can obtain only a limited number of data from the experiments, which is insufficient to meet the requirements for training an effective classification and recognition model. In this paper, we start with an existing data augmentation method called DBA (for dynamic time warping barycenter averaging) and propose a new data enhancement method called AWDBA (adaptive weighting DBA). We first validated the proposed method by synthesizing new data from insulator sample datasets. The results show that the AWDBA proposed in this study has significant advantages relative to DBA in terms of data enhancement. Then, we used AWDBA and two other data augmentation methods to synthetically generate new data on the original dataset of insulators. Moreover, we compared the performance of different machine learning algorithms for insulator health diagnosis on the dataset with and without data augmentation. In the SVM algorithm especially, we propose a new parameter optimization method based on GA (genetic algorithm). The final results show that the use of the data augmentation method can significantly improve the accuracy of insulator defect identification.

## 1. Introduction

Insulators play an important role in maintaining the reliability and safety of transmission and distribution systems [1,2]. However, insulators often fail due to their own defects (e.g., crack, voids, etc.), which can result in power grid failure and lead to enormous economic losses [3,4,5]. Therefore, it is very necessary to assess the health condition of insulators through nondestructive testing methods. Among many nondestructive testing methods, the ultrasonic testing is widely used for defect detection owing to its advantage of high speed and sensitivity [6]. However, there are some drawbacks for this approach. For one, in ultrasonic testing, correct identification of the health condition of insulator is determined largely by the experience of technicians to interpret the 1D ultrasonic time series signal. In some cases, it is extremely difficult for engineers to access the real insulator condition. For another, this method is prohibitively costly, time-consuming, and unsafe. To address these issues, using the machine learning (ML) method to assist workers in better and more rapidly diagnosing the health of insulators is a smart concept [7].

Unfortunately, due to the impact of detection cost and a hazardous environment, collecting enough data under an actual working environment is extremely difficult, and a small set of data may not meet the needs of ML algorithms to train a high classification accuracy model. In this instance, using data augmentation (DA) techniques to create new copies of data can enlarge the dataset size so that the precision of the model classification can be improved [8]. Data augmentation is a strategy to improve the generalization capabilities of trained models by changing current samples into new data to avoid overfitting of the model [9,10]. However, despite the widespread use of data augmentation approaches in computer vision [11] and speech recognition [12], data augmentation in time series needs further study [10]. Many early strategies for augmenting time-series data were borrowed from picture data augmentation [13] (e.g., cropping, flipping, noise addition, et al.). These methods commonly rely on random modifications of the original dataset to generate new data. However, not all transforms are appropriate for all datasets [14]. For example, the noise addition might be true for sensor and audio data, but this is abnormal for time series based on object contours. Another example is frequency warping, which is not available for audio datasets.

Different from the earlier work, some new approaches have been proposed in recent years, such as the dynamic time warping (DTW) barycenter averaging (DBA), which creates a new time series based on a sequence set [15]. The DBA method involves averaging a time series set base on DTW to synthesize a new sequence that can mix the characteristics of various sequences in the dataset, so the new sample can better represent the global features of the sequence set. Researchers have conducted many studies after DBA was first used for data augmentation. Based on [15], Forestier et al. [16] proposed a weighted version of DBA (WDBA), and three weighting schemes: average all (AA), average selected (AS), and average selected with distance (ASD) given. In recent years, DAB and WDBA have been applied by several researchers in sequence classification and regression problems [8,17,18,19]. For example, Manabu Okawa [19] used the DBA to improve the rate of online signature verification. Forestier et al. [16] and Fawaz et al. [17] used the WDBA for time-series classification on different datasets, and the results showed that ASD and AS are superior to AA. Unfortunately, there are still some shortcomings with ASD and AA. The first disadvantage is that the assignment of each time-series weight depends on the one closest to the initial average sequence, and this weight value is given by the researcher (such as 0.5 in ASD). In other words, the weights of some sequences (e.g., the closest to the initial one) are assigned manually and not adaptively based on each time series. Although these weighting schemes seem to provide a good result, it is difficult for a user to give a suitable weight value for all the sequence set. For another, the WDBA technique for creating new time series involves using an initial average sequence to update and iterate constantly, and this initial sequence is determined by averaging the entire sequence set. However, if the sequence set includes some anomalous samples, an abnormal initial sequence may arise, which will be harmful for the next iteration. These issues are discussed in detail in Section 2 and Section 3.

Based on the above issues, in this paper, a novel adaptive weighting method is proposed to overcome the limitations of classical WDBA, called adaptive weighting DBA (AWDBA). In AWDBA, the weight is defined by the distance between every sequence and the initial average series. Moreover, a new iteration strategy is adopted to avoid the influence of the abnormal sample. The paper is organized as follows: Section 2 introduces some related works. Section 3 discusses in detail the AWDBA and new iteration strategy. A few ML algorithms and other DA technologies used in this article are described in Section 4. In Section 5, we introduce the data collection, feature extraction of data, and training strategy and performance measures of machine learning algorithms. Then, in Section 6, we discuss the impact of difference augmentation techniques on the performance of different classification algorithms to verify the importance of data augmentation for machine learning to achieve better classification performance. Finally, the Conclusions and Future Work section is presented in Section 7.

## 2. Related Work

Many DA technologies have been proposed over the past decade [14]; however, we mainly focused on the DBA method and modified it to augment the time-series signal. Thus, in this section, we discuss the related work in the areas of DA and DBA techniques for time-series analysis.

### 2.1. The Study of Time-Series Augmentation

The purpose of data augmentation methods is to generate new samples, which include the unconsidered feature space of all classes, in order to improve classification performance [20]. Some simple approaches, such as jittering, scaling, rotation [13,21], numerical simulation [9], and magnitude warping [10] produce a new sequence relying on random transformations of the size of signal magnitude. Other techniques involve window slicing [22] and time warping [10] which are implemented by changing the time steps of the sequence. Rashid et al. [23] employed this strategy to increase the construction equipment activity dataset. Meanwhile, an RNN (recurrent neural network)-based long short-term memory (LSTM) network is trained to predict the condition of the equipment. The research shows that the introduction of data augmentation helps to reduce interclass confusion in the LSTM network. Recently, DNN-based generative adversarial networks (GANs [24]) have also received considerable attention in the field of DA. Lou et al. [25] incorporated an auto-encoder network with a fully connected network (WGAN) to obtain more new data. Harada et al. [26] and Esteban et al. [27] directly used deep LSTM-based recurrent GANs to augment medical time-series signals. Differently from the methods mentioned above, Giorgia et al. [28] proposed the temporal convolutional GAN (T-CGAN) method. It uses the 1D CNN for time-series generation. In addition, there are many GAN models used to synthesize new data for different classification recognition tasks. In the electronic health records area, Che et al. [29] developed the ehrGAN, which uses a new encoder with variational contrastive divergence in generative networks. Chen et al. [30] presented emotional GAN grounded in 1D CNNs for classifying emotions from long ECG patterns. The accuracy of classifying emotions was notably improved after augmentation with data techniques. Table 1 presents a comparison of some common data augmentation methods for time series.

### 2.2. DTW Barycenter Average (DBA) for Data Augmentation

As another important DA approach, the DBA was first proposed in [15]. Here, we give a brief overview of dynamic time warping (DTW) before introducing the DBA. 

DTW was first introduced in the 1970s for speech recognition [31,32,33]. It tends to be used to determine the similarity of two sequences and to find a nonlinear alignment optimal path by minimizing the cumulative distance between $X(x_{1},\ldots ,x_{i},\ldots ,x_{T})$ and $Y(y_{1},\ldots ,y_{i},\ldots ,y_{T})$ [34,35,36]. The distance between elements of *X* and *Y* is defined as
(1)$$d(x_{i},y_{j})={\left(x_{i}-y_{j}\right)}^{2};1\leq i\leq T,1\leq j\leq T$$
where *T* is the length of sequence *X* and *Y*. We can obtain a cumulative distance matrix *D* through these two sequences (row and columns), where the element D(i,j) is computed with
(2)$$D(i,j)=d(x_{i},y_{j})+\min\left\{ \begin{array}{l} D(i,j-1)\\ D(i-1,j-1)\\ D(i-1,j) \end{array}\right.;i,j=1,..,T$$

DTW provides a way to measure the similarity of two sequences. Moreover, it also defines an optimal warping path between two signals. This path aligns numerical values of two sequences to increase their similarity. An example of DTW for aligning two out-of-phase time series is shown in Figure 1.

However, DTW does not account for the significance regarding the phase difference between the two aligned sequence points. This may lead to misclassification, especially in cases where the shape similarity between two sequences is a very important factor for accurate recognition. To overcome this problem, Young-Seon et al. [37] proposed a novel distance measure method, called a weighted DTW (WDTW). The WDTW penalizes points with a higher phase difference between the two aligned sequence points to obstruct the minimum distance distortion caused by outliers. The experimental results indicate that WDTW can remarkably improve accuracy for time-series classification. Furthermore, in order to solve the problems of averaging multiple time-series methods, Francois et al. [15] designed a global barycenter average strategy based on DTW (DBA) to acquire the average time series. The main steps of the method are as follows: **Step1**:Randomly choose an initial average time series ${\bar{S}}_{init}$ from the sequence set $X=\{X_{1},\ldots ,X_{i},\ldots ,X_{M}\}$, where *M* is the number of the sequence in *X*.**Step2**:Align every $X_{i}$ on ${\bar{S}}_{init}$ and compute the warping paths.**Step3**:According to the path, obtain the new coordinates of each $x_{i}$ and update the initial average sequence by averaging their coordinates.

Algorithm 1 describes the process of DBA in detail, and Figure 2 illustrates the process of generating an average signal on the two examples of Figure 1 using the DBA.
**Algorithm 1.** DTW barycenter average (DBA)**Require:** an initial average sequence: ${\bar{S}}_{init}=(s_{1},\ldots ,s_{i},\ldots ,s_{T})$
**Require:** the sequences set $X=(X_{1},\ldots ,X_{i},\ldots ,X_{M})$
    where $X_{i}=(x_{1},x_{2},\ldots ,x_{T})$
**Let**
*T* be the length of the initial average sequence
**Let**
*M* be number of the sequence set *X*
**Let**
D[i,j] be a temporary *DTW* (distance, path) matrix
where D[i,j,1]←DTW(dis) and D[i,j,2]←DTW(path)
**Let**
coorTab be a table of size *T* with a set of coordinates in each cell corresponding to each $s_{i}$ coordinate
  coorTab[k]←Φ, k=1,2,…,T
**for**
1≤t≤M
**do**
  $D=DTW(S,X_{t})$
  **for**
1≤i≤T;1≤j≤T
**do**
    $coorTab[i]\leftarrow coorTab[i]\cup x_{j}$; [i,j]←D[i,j,2]
  **end for**
**end for**
**for**
i=1
**to**
*T*
**do**
  $\bar{S}\leftarrow mean(coorTab[i])$
**end for**
**return**
$\bar{S}$

Since DBA was first presented, many studies involving the time series have been conducted. Petitjean et al. [38] synthetized representative time series from a group of data with DBA technology, which can simplify the dataset to improve the operation rate of the algorithm. Meanwhile, one of the main limitations of DTW and DBA occurs when a point of $s_{i}$ is matched to several points of $x_{i}$. This is when a path is considered to “stagnate” or become a “pathological path”. To overcome this problem, Marion Morel et al. [18] proposed an alternative method based on a constrained dynamic time warping (CDBA) to avoid the problem of pathological warping paths and improve the global model recognition rates. In the online signature verification field, Manabu Okawa [39] modified DBA and constructed a novel time-series averaging method called Euclidean barycenter-based DTW barycenter averaging (EB-DBA) to address the problem of the lower performance and higher calculation complexity of online signature verification. In [19], he further combines the EB-DBA and CDTW to simultaneously increase the speed and accuracy of online signature verification. In particular, a key point worth noting is that the DBA can not only be used for time-series analysis, but for data augmentation. Forestier et al. first used DBA for data augmentation in [16]. Oszust [19] proposed a new data augmentation method for an action recognition dataset, in which the DBA is used to remove potential new time series that do not offer additional knowledge to the examples of a class or are created far from the occupied area. In their study, the authors utilized a new WDBA approach to synthesize average time series as augmentation data. In WDBA, each element of the sequence set is not viewed equally, and some time series contribute more to the final average result than do others. It weights each sequence and uses the weighted arithmetic mean to compute infinitely more additional values as the augmentation data. Algorithm 2 shows the detailed process of using WDBA to generate novel time series.
**Algorithm 2.** WDBA: Weighted _ DBA (X, ${\bar{S}}_{init}$, W, I)**Input:** the set of sequences $X=(X_{1},\ldots ,X_{i},\ldots ,X_{M})$
   where $X_{i}=(x_{1},x_{2},\ldots ,x_{T})$
**Input:**
${\bar{S}}_{init}$: the initial sequence (of length *T*)
**Input:**
*W*: the weights set of time series X, where $W=(W_{\left(1\right)},\ldots ,W_{\left(i\right)},\ldots ,W_{\left(M\right)})$
**Input:**
*I*: the number of iterations
**Output:**
S: a table of size *I* and each cell matches a new data
      S[k]←Φ, k=1,2,…,I
**Let**
$\bar{S}=[0,0,\ldots ,0]$ be a length of *L*
**for**
1≤i≤I
**do**
 **for**
1≤t≤M
**do**
    $D=DTW(S,X_{t})$
     **for**
1≤i≤T;1≤j≤T
**do**
      $coorTab[i]\leftarrow coorTab[i]\cup x_{j}$      {see **Algorithm 1**}
      $\bar{S}(i)\leftarrow \bar{S}(i)+coorTab[i]*W_{\left(i\right)}$
      [i,j]←D[i,j,2]            {see **Algorithm 1**}
     **end for**
 **end for**
 **for**
i=1
**to**
*T*
**do**
  $\bar{S}(i)\leftarrow \bar{S}(i)/\sum_{i=1}^{M}W_{\left(i\right)}$
 **end for**
  ${\bar{S}}_{init}\leftarrow \bar{S}$; $S[k]\leftarrow S[k]\cup \bar{S}$
**end for**
**return**
S

For WDBA, two key points should be noted: (1) what value the initial sequence ${\bar{S}}_{init}$ should be allocated, and (2) how the weight $W_{i}$ of each time series is calculated. 

Regarding the first problem, it can be solved by two methods: randomly select $X_{i}$ from the dataset *X* as the initial sequence ${\bar{S}}_{init}$ or average the entire set of sequences. Forestier et al. seem to be suggest that the first approach is preferable in their papers [15,16]; however, if an anomaly sample in the dataset happens to be selected, there is no doubt that it will have a considerably negative impact on the final result. Furthermore, according to their work, the previous result ($\bar{S}$) is used as the next initial sequence (${\bar{S}}_{init}$) for iteration calculation. It is easy to see that if the preceding result $\bar{S}$ is an anomalous sample, the next average result ${\bar{S}}_{init}$ will be influenced after completion of an iteration, which may lead to many anomalous data. As for the second problem, Forestier et al. designed three solutions (AA, AS, ASD) to calculate the weight. Forestier et al. [16] and Bandara et al. [8] hold that ASD technology is more suitable for DBA to augment data. The works of Fawaz et al. [17] demonstrated that the DBA using AS also had a good performance. Regrettably, whether AA or ASD is used, the values of weight are assigned according to the author’s experience. For example, the AS starts by randomly picking a sequence $X_{i}$ and giving it a weight of 0.5. Then, choose 2 from the 5 nearest neighbors of $X_{i}$ and assign them a weight of 0.15. Finally, uniformly assign the remaining 20% to the rest of the time series. Differently from AS, the ASD (Equation (3)) takes into account the relative distance between the initial time series and the other sequences, but it still gives the sequence closest to ${\bar{S}}_{init}$ a 0.5 weight. Obviously, this is an empirical value and not a generalized conclusion.
(3)$$W_{i}=e^{\ln(0.5)*\frac{DTW({\bar{S}}_{init},X_{i})}{d*(NN)}}$$
where d*(NN) is the distance between ${\bar{S}}_{init}$ and its nearest neighbor.

## 3. Proposed New Method for Data Augmentation

In order to address these difficulties of existing WDBA, the following aspects should be considered.The initial sequence ${\bar{S}}_{init}$ cannot simply be selected from the dataset.Instead of using the previous result $\bar{S}$ as the next initial sequence ${\bar{S}}_{init}$ to conduct the next calculation, the iteration strategy of producing new data should be adjusted.The weight values of every sequence should be calculated according to the relative between them and the initial sequence.

In next section, we will discuss these points and introduce a new method called AWDBA for data augmentation.

### 3.1. The Initial Time Series and Iteration Strategy

As noted previously, a few anomalous samples, which are invariably gathered in experiments, have a severe impact on the final average results, particularly when a bad sample is chosen as the initial sequence. Some methods have been proposed to address problems of abnormal samples, such as preprocessing methods for outlier removal in time series [40]. In this paper, we introduce a new iteration strategy to reduce the impact of an abnormal sample to a new synthetic sample. Firstly, we select the second scheme, which averages the entire set of sequences as the initial sequence ${\bar{S}}_{init}$. The reasons for this are as follows: the results of averaging the sequence set are nearer to the right sample than are those produced by randomly selecting a sample from the series set (while the normal sample may be chosen, it is a probabilistic event) because the normal time series significantly outnumbers the anomaly time series in a dataset. For sequence set *X*, the initial time series ${\bar{S}}_{init}$ is calculated with
(4)$${\bar{S}}_{init}=\sum_{i=1}^{M}X_{i}/M$$
(5)$${\bar{S}}_{init}(k)=\sum_{i=1}^{M}X_{i}(k)/M$$
where ${\bar{S}}_{init}=\{{\bar{S}}_{init}(1),\ldots ,{\bar{S}}_{init}(k),\ldots ,{\bar{S}}_{init}(T)\}$, $X_{i}=(x_{1},\ldots ,x_{k},\ldots ,x_{T})$, and $X_{i}(k)=x_{k};1\leq k\leq T$.

In addition, if the initial sequence is abnormal, the first average result, as with the second initial sequence, will have a negative impact on the second average result, which requires us to modify the iteration strategy. A good way to do this is to add the average sequence of the previous calculations to the sequence set to be averaged in the next step. The reasons for this idea are as follows. Firstly, the new data is closer to the normal sample because we changed the methods of choosing the initial sequence. Secondly, if we add the data to the set of sequences, the proportion of healthy samples among all data will increase, which reduces the influence of the anomaly sample on the final results in the next iteration. This strategy is described as follows:(6)$$X^{n+1}=X^{n}\cup {\bar{S}}^{n};(n\geq 1)$$
where, $X^{n+1}$ is the sequence set used to average for the (N + 1)th time 

$X^{n}$ is the sequence set used to average for the Nth time, and

${\bar{S}}^{n}$ is the average sequence generated for the Nth time. 

The new method of selecting the initial sequence and the iteration strategy avoids the impact of abnormal data on augmentation data, but the weighting question is still unsolved; it will be discussed in the next section.

### 3.2. Adaptive Weighting

It is common sense that the weight of each time series should consider the characteristics of the sequence itself, and the ASD seems to do this. However, as we discussed before, it is still somewhat incomplete. Returning to DTW, we define a novel approach to calculate weight. DTW can measure the similarity between two sequences by their final cumulative distance. For instance, if the distance between A and B is greater than A and C, we believe that A is more similar to C than B. In other words, if a sequence is closer to the initial average sequence, then they correspond to a smaller final cumulative distance and should be given a larger value of weight. We begin our discussion from these concepts. Firstly, let $D_{i}$ be the final cumulative distance between $X_{i}$ and ${\bar{S}}_{init}$, and it is described as
(7)$$D_{i}=\mathrm{DTW}(X_{i},{\bar{S}}_{init})$$
where $X_{i}$ is from the set of sequences $X=(X_{1},\ldots ,X_{i},\ldots ,X_{M})$, and

${\bar{S}}_{init}$ is the initial average sequence.

Then, we normalize the distance and normalization factor $\eta_{i}$, which is defined as
(8)$$\eta_{i}=\frac{D_{i}}{\sum_{i=1}^{M}D_{i}};\eta_{i}\in (0,1)$$

From Equation (8), if $X_{i}$ is nearest ${\bar{S}}_{init}$, a smaller $\eta_{i}$ is obtained, and yet we want $X_{i}$ to be assigned a higher weight value when it is more similar to ${\bar{S}}_{init}$. Thus, the final value of weight function $W_{\left(i\right)}$ is defined as follow:(9)$$W_{\left(i\right)}=e^{-\eta_{i}}=e^{-\frac{D_{i}}{\sum_{i=1}^{M}D_{i}}}$$
where *I =* 1, 2, …, *M*.

Finally, the weight value for each sequence can be obtained by the value of each weight function divided by that of the sum of all weight functions. In contrast with AA and ASD, adaptive weight (AW) does not need to set the empirical value manually, and it provides the weight value by the relationship between two sequences to prevent the generation of unusual data. Algorithm 3 details the process of data augmentation using AWDBA.
**Algorithm 3.** Adaptive Weighting _ DBA (X, ${\bar{S}}_{init}$, W, I)**Input:** the set of sequences used to average $X=(X_{1},\ldots X_{i},\ldots ,X_{M})$
**Input:**
${\bar{S}}_{init}=\sum_{i=1}^{M}X_{i}/M$
**Input:**
$W_{\left(i\right)}=\exp(-D_{i}/\sum_{i=1}^{M}D_{i})$          {$D_{i}$ see Equation (9)}
**Input:**
*I*: the number of iterations
**Output:**
S: a table of size *I* and each cell match a new dataset
      S[k]←Φ, k=1,2,…,I
**Let**
$\bar{S}=[0,0,\ldots ,0]$ be a length of *L*
**for**
1≤i≤I
**do**
 **for**
1≤t≤M
**do**
   $D=DTW(S,X_{t})$
   **for**
1≤i≤T;1≤j≤T
**do**
      $coorTab[i]\leftarrow coorTab[i]\cup x_{j}$       {see **Algorithm 1**}
      $\bar{S}(i)\leftarrow \bar{S}(i)+coorTab[i]*W_{\left(i\right)}$
      [i,j]←D[i,j,2]              {see **Algorithm 1**}
    **end for**
 **end for**
 **for**
i=1
**to**
*T*
**do**
   $\bar{S}(i)\leftarrow \bar{S}(i)/\sum_{i=1}^{M}W_{\left(i\right)}$
 **end for**
  $S[k]\leftarrow S[k]\cup \bar{S}$
 $X^{k+1}=X^{k}\cup \bar{S}$; $X^{1}=X$              {see Equation (6)}
**end for**
**return**
S

Briefly, this new method of data augmentation does the following: It takes the average value of the sequence set as the initial sequence to synthesize the first new sequence (the first augmentation data) by AWDBA. It then adds the new data to the raw data to generate a new initial sequence by averaging the new dataset. Finally, it uses the new initial sequence to synthesize the second sequence by AWDBA again and cycle in turn. With AWDBA, we can set the iteration number (*I*) to augment the same amount of data.

### 3.3. Evaluation

In this section, we compare the performance of AWDBA and WDBA (with ASD) on the synthetic average time series in the insulator dataset (the details of the dataset are shown in Section 5.1). According to Algorithm 3, each of iteration synthesizes a new sequence. Here, the original dataset of each insulator specimen is used for data augmentation with AWDBA and WDBA, respectively. To quantify the similarity between the new augmentation data and the original dataset, a new variable called average distance (*AVG**_Dis*) is used to measure the similarity of the sequence set, and new data are introduced. Let *AVG_Dis* be the distance between new sequence $\bar{S}$ and the sequence set *X*, and then obtain the average, which is defined as
(10)$$AVG\_Dis=\frac{1}{M}\sum_{i=1}^{M}Dis(\bar{S},X_{i})$$
(11)$$Dis(\bar{S},X_{i})=\sqrt{\sum_{i,j=1}^{T}d(s_{i},x_{j})}$$
where *M* is the number of the sequence set *X*.

*AVG**_Dis* is the average distance between the new sequence and the original dataset, and a smaller average distance indicates that the new data are very similar to the original dataset. As shown in Figure 3, the average distance between the new data and the original data set synthesized by WDBA increases with the number of iterations, which indicates that the new samples of the latter are less similar to the original data. By contrast, the average distance of all new data of every specimen synthesized by (AWDBA) is smaller than the WDBA, which indicates that the data generated by AWDBA are closer to the raw sample.

Figure 4 shows the results of some of the data augmentation; it can be observed that many synthetic signals obtained by WDBA lose the feature of most normal data. This is because the initial sequence is an anomalous sample that was randomly selected and given a larger weight by WDBA. 

Table 2 shows the average *AVG_Dis* for all augmentation data of each specimen. From these results, a conclusion can be drawn that adopting the adaptive weight and novel iteration strategy to optimize the DBA method can obtain some data that have high similarity with the raw time series.

## 4. Other DA Techniques and Machine Learning Algorithms

In the previous section, we describe the AWDBA algorithm and demonstrate that it has more advantages than does the WDBA for data augmentation. In this section, we introduce other DA techniques, which can be used with AWDBA to enlarge the scale of the insulator dataset. Additionally, a few machine learning classification algorithms (e.g., GA-SVM, KNN, logistic regression, decision trees, and random forests) that are used to diagnose the health of insulators are briefly summarized.

### 4.1. Data Augmentation Techniques

As described in Section 2.1, although image augmentation can be used as a reference for time series, some methods may not be applicable for time series data. For example, the flipping method is taking the opposite value of the magnitude of the sequence. However, it is not different from the original signal when the amplitude takes the absolute value in the calculation. Meanwhile, in practice testing, some factors such as sensor placements and temporal characteristics of activities often cause slight changes in the original signal [23]. In order to account for these factors and obtain the value of change in the original signal, slicing and scaling, as shown in Figure 5 [14], are often used. In this article, except for AWDBA, we also use these two methods for data augmentation. The following is a short introduction to these two methods:

*Slicing:* Slicing is also sometimes referred to window slicing (WS) [22]. It samples the time series through a moving window and then stretches the new sampling data to the length of the original sequence. A new sample is described as
$$X_{i}{}^{\prime }=(x_{\varphi },\ldots ,x_{k},\ldots ,x_{\varphi +w})1\leq \varphi \leq T-w$$
where φ is a random integer such as 1≤φ≤T−w and w is a size of a window. Figure 5a shows an augmented dataset after slicing was applied to the raw time series. In this paper, the size of window is set to 3.

*Scaling*: Scaling is another important technique used in data augmentation, which changes the magnitude of the raw data but preserves the labels. During signal acquisition, the global magnitude of a time series changes with the change of the installation sensor tool. Scaling generates new signal data by using a random number α to multiply the amplitude of the raw signal. For $X_{i}$, scaling is a multiplication of $\alpha_{i}\left(i=1,\ldots T\right)$ to the entire time series: $$X_{i}{}^{\prime }=(\alpha_{1}x_{1},\ldots ,\alpha_{K}x_{k},\ldots ,\alpha_{T}x_{T})$$
where the parameter $\alpha_{i}\left(i=1,\ldots T\right)$ can be determined by a Gaussian distribution $\alpha \sim N(1,\sigma^{2})$ with σ as a hyperparameter [10]. In this paper, we let σ equal 0.08. The effect of scaling on the dataset is illustrated in Figure 5b. 

### 4.2. Machine Learning Algorithms

Machine learning can be utilized to address either a regression or a classification problem, and the insulator condition diagnostics can be seen as binary classification issues. In recent years, many different classification algorithms have been applied to monitor the healthy condition of insulators. Examples of these classifiers include logistic regression, k-nearest neighbors (KNNs), support vector machines (SVMs), decision trees, and random forests. The result of using these machine learning algorithms revealed that there is no single algorithm that consistently outperforms other algorithms, so we compared the classification performance of different algorithms, as outlined in the experiment section. Now, we give a short explanation of these algorithms.

*Logistic Regression (LgR):* The LgR method is invented from line regression (LR), where the model output is y∈[0,1], and y represents the probability that the input x belongs to one of two possible categories. Usually we take 0.5 as a threshold value, where (y>0.5,x=1;y<0.5,x=0). The LgR is mainly designed to address a binary classification problem, and many examples reveal that the LgR has good performance for the classification problems with a smaller dataset. 

*K-Nearest Neighbor (KNN):* The KNN classifier considers that the train samples are vectors in Euclidean spaces, and the labels can be generated automatically by analyzing the relationship between each of them. During classifier operation, the samples of similar features will cluster a class by Euclidean distance and form a sample center. The final number of the class is determined by setting a hyperparameter *K*. In the prediction phase, if the input sample is nearer to the center of the *K^th^* class sample, it is considered to belong to the *K^th^* class. 

*Support**Vector**Machine**With Genetic Algorithm (**GA_SVM)**:* The support vector machine is another important approach to binary classification problems that attempts to find a hyperplane that represents the margin between the two classes, with labels −1 and 1. However, the data tends not to be linearly separable in many classification tasks, and so the SVM applies kernel functions where the training samples are mapped onto a higher-dimensional feature space, where the dataset may be classic. There are many different kernel functions in SVM, but the most frequently used is the Gaussian kernel function (RBF kernel): $$RBF\;kernel:K(x_{i},x_{j})=\exp(-g{\left\|x_{i}-x_{j}\right\|}^{2})$$
where *g* is a hyperparameter of the RBF kernel function. The performance of the SVM classifier has a crucial impact if a more significant *g* is given, which is likely to cause underfitting. On the contrary, the smaller the *g* is worse for the susceptibility of overfitting and the generalization ability of the model. The key for SVM is to find a suitable value of *g*. In addition, the concept of regularization is often used in SVM to avoid misclassifying the training samples, and the penalty parameter *C* is set to control the smooth decision boundary and to classify the training points correctly. In summary, the SVM model requires the tuning of hyperparameters, and the *GridSearch* [41] is the most common method. However, *GridSearch* has low efficiency and is time-consuming when the search interval of *g* and *C* is very large. Compared with the *GridSearch*, the genetic algorithm (GA) has the characteristics of fast and global optimization. However, the GA needs an initial parameter search interval, and this interval is larger. If we want to acquire more accurate results, a larger population size should be used. This is still time-consuming.

Thus, in this paper, a parameter optimization strategy [42] is used by combining the *GridSearch* method with GA. How this strategy works is shown in Figure 6. We first search the optimal value of *C**** and *g** by using the GA method, where *C* and *g* are taken in larger random intervals. Then, we use *GridSearch* to find optimum values *C**^best^* and *g**^best^* in a small range determined by *C** and *g****. This strategy takes advantage of the fast search capability of GA and avoids excessive computation.

*Decision Tree**(DT) and Random Forest**(RF):* Different from other classification methods, decision trees involve recursive thinking and use binary decision rules to cut off the feature of the training sample that partitions the dataset into *N* disjoint sets $S_{1},\ldots ,S_{k},\ldots ,S_{N}$. If a new input sample $x\in S_{k}$, then the prediction value of the class is *k*. The decision tree algorithm has many versions. The *C4.5* algorithm is used in our work. The main limitation of decision trees is that they are prone to overfitting by creating overcomplicated models that view the feature of the training set as all data characteristics [43]. The random forest can avoid this problem. We can train individual decision trees to construct a forest, in which the training sample of each tree selects from a raw dataset in proportion (such as 70%). In parallel, the data of each tree do not include all features but select some features proportionally (such as 80%). Finally, many prediction results are output from different decision trees, and we select the most frequent result as the final prediction value of the random forest. 

## 5. Experimental Study

In this section, we collected data using ultrasonic nondestructive testing for insulators. Then, AWDBA and the other two DA methods were used to augment the data. We describe the extraction method of the data feature, the performance measures of the algorithms, the training strategy of the model, and the computational capabilities used in our experimental setup. Finally, we provide a detailed analysis of the results obtained.

### 5.1. Data Collection

Considering the variety of the outdoor insulator, we collected data from three insulator specimens with an ultrasonic detector (OND CT-50). The first was the reference specimen (Specimen 1), a porcelain pillar with only surface cracks and no internal flaws, which it is often used to simulate the surface defects of the strut porcelain insulator. The second was the standard specimen (Specimen 2), which was used to set the insulator testing standards. It is made of aluminum alloy with a similar sound speed to porcelain insulators and contains artificial defects, such as surface cracks and internal voids. The third was an outdoor insulator (Specimen 3) that was removed from the power grid and selected for flaw detection. We confirmed that it had internal defects through industrial CT (Y. CT Modular). 

The ultrasonic nondestructive testing experiment system consisted of an ultrasonic fault detector, a router, insulator specimens, a sensor, and a computer, as shown in Figure 7. When an ultrasonic fault detector is used for nondestructive testing, the probe connected to the instrument is placed on the surface of the specimen, and the signal is transmitted to the fault detector and sent to the computer via a router at the same time. In addition, it is necessary to use a coupling agent (“*cofoe*”—medical ultrasonic coupling) between the probe and the surface of the specimen to ensure the stability of the signal. Table 3 summarizes the total data from our experiments. Here, 78, 164, and 90 datasets were collected separately from Specimen 1, Specimen 2 and Specimen 3. The healthy and defective datasets gathered were 131 and 201, which were marked by 0 and 1 respectively. Following data collection, all data were utilized to execute augmentation techniques (which include AWDBA, scaling, and slicing), with each augmentation technique generating 1-fold augmented training data. Thus, the final data set was four times the size of the original data set. Finally, we used principal component analysis (PCA) to extract key features from the 1328 instances.

### 5.2. Feature Extraction and Selection

It is not appropriate to use ultrasonic signals directly as the feature data for machine learning because it may decrease the accuracy of the machine learning analysis. Therefore, it is necessary to apply a suitable method to reduce the amount of data and retain key feature of data. However, using traditional artificial extraction algorithms (e.g., the lower area of the curve, the root mean, the median of the curve, etc.) to obtain the most representative features is difficult. Here, we employed PCA for feature extraction since the data categories are known. PCA is one of the most important tools for dimensionality reduction, data processing, compression and key feature extraction, and pattern detection [44]. The purpose of PCA is to identify *N* orthogonal axes with the largest variance in data and then project the original data to the *N* principal direction to generate new *N* dimensional samples that are much smaller than the original sample size, so as to better characterize the original data and improve the classification performance. 

In machine learning, we may not train a classifier with the high accuracy if we use all feature data as the input data. One possible reason for this is that the raw features do not contain the presence of damage or have redundant information. Therefore, the feature selection tends to select the most relevant ones from raw features [45]. In addition, the feature selection technique can also reduce the dimension of input data. There are two main methods in feature selection: filter [46] and wrapper methods [47]. The wrapper methods search for the optimal number of features according to the classification accuracy of a learning algorithm. First, the dataset is divided into many subsets with different feature combinations. We then test each subset and find the optimal subset as the final choice. However, these methods are computationally expensive. Therefore, many heuristics have been proposed in some studies. For example, in Ref [45], the author uses the sequential floating forward selection (SFFS) method to select the most relevant 3 features of 8 features extracted from the time domain signals and confirm the sensibility of the selected feature. 

In this paper, feature extraction is performed by PCA. We reduce the dimensions of the raw signal from 501 to 20 as the feature of the sample (Feature 1 to 20). These features are fed to the SFFS method for feature selection. The algorithm begins by testing each feature individually and then selecting the one that gives the higher accuracy of classification. The K nearest neighbor (KNN) method was used as a criterion for the estimation of this accuracy. The result in Figure 8 shows that, from twenty extracted features, only three were selected: Feature 1, Feature 2, and Feature 3.

### 5.3. Performance Measure and Training Strategy

In order to compare the performance of classification algorithms before and after data augmentation, the confusion matrix was used to evaluate the models by computing the accuracy, precision, recall (or sensitivity), and F1 score. Accuracy can be calculated by dividing the number of correct predictions by the total number of classes. Precision is the true-positive part of a positive example of a prediction, and recall refers to the proportion of true-positive classes that are correctly predicted to be positive. F1 score is defined by the weighted harmonic mean between precision and recall. 

In addition to these standard metrics from the confusion matrix, we examined the mean square error (MSE) of the model, which was used in the random forest and GA_SVM: (12)$$\mathrm{MSE}=\frac{1}{N}\sum_{i=1}^{N}[f(x_{i})-y{]}^{2}$$
where *N* is the testing set size, $f(x_{i})$ is the predicted value of the sample, and *y* is the actual value. 

Before training the model, in order to decrease the error resulting from large data variation, the signal amplitude is normalized by Equation (13): (13)$$\bar{X}=\frac{2\times (X-X_{\min})}{X_{\max}-X_{\min}}-1$$
where $X_{\max}$ and $X_{\min}$ are the maximum and minimum values of the data.

In another aspect, if the data are simply divided into a training set and testing set, it is difficult to train a model with high generalization ability because the prediction accuracy may decrease with the change of training and test data. Thus, the 10-fold cross-validation method was used in this paper. The dataset is split into ten equal parts, with each part consisting of the defects and healthy samples (labels 1 and 0) of the same proportion, and we selected the first part as the testing set to estimate the performance of the model and the remaining nine as the training set to train the model. In other words, during a training round, the training and testing data accounted for 90% and 10% of the total data, respectively, and this process was performed ten times. Finally, every measure index had 10 values, and we averaged the total results as the output value of the metric.

### 5.4. Computational Performance

A brief description of computer performance is given in this section. Our experimental calculations were run on an Intel(R) i7-10700 processor (2.90 GHz), with 2 threads per core, 8 cores in total, and 16 GB of main memory.

## 6. Results and Discussion

### 6.1. Impact of Data Augmentation Type

In this section, we compare the classification performances of the AWDBA with the DBA (weight is given by AA, AS, and ASD) and other augmentation techniques (scaling and slicing) on the results are shown in Figure 9 and Figure 10. The comparison of results showed that the AWDBA had the better performance in each algorithm than did the DBA, scaling, and slicing. It is important to note that we only performed 1-fold data augmentation on the dataset by each technique. Therefore, the size of training data for each data augmentation approach was twice as big as that of the raw dataset.

### 6.2. Performance of Data Augmentation vs. No Data Augmentation

In this subsection, we show the results of different machine learning algorithms for insulator health diagnosis and discuss whether the data augmentation techniques improved the classification performance of the model. The gradient descent approach was used to optimize the parameter for the logistic classification algorithm, and the learning rate α given by 0.005. Then, we ran 100 training pieces, each using a 10-fold cross-validation method. Table 4 gives the maximum value of four indexes in 100 training results (accuracy, precision, recall, and F1 score) with data augmentation. As seen from these results, the prediction accuracy of the algorithm has a remarkable increase after the DA techniques are applied. 

The three points to note in KNN classifier are the number of nearest neighbors (K), the distance function, and the weight function of distance. As mentioned in Section 4.2, the value of *K* is set from 1 to 10. In this paper, the Euclidean distances were computed as the distance function. In addition, the inverse of distances was used as a weight function. Figure 11 and Table 4 show these results. It can be seen that the data augmentation also improves the performance of the KNN classifier and that the accuracy of the KNN algorithm (0.926) is higher than that of the LgR algorithm (0.636). 

For GA_SVM, as Figure 6 shows, we need to build a population of several individuals that include two gene sequences of a particular length or a chromosome. The population produces offspring through a certain probability of breeding and gene mutation. Then, the binary code of each offspring individual is used to represent the *C* and *g* in a given interval. Thus, many different SVM models are generated, and we constantly repopulate and select until finding the optimal one (also optimal *C* and *g*). Here, the objective function of the GA is the MSE (Equation (12)) value of a different model. Finally, *GridSearch* is used to search for a final result in a small range based on the optimal *C* and *g* values by GA. Table 5 provides the information of several parameter settings of the GA, including the iterations, population size, gene mutation probability, breeding probability, and the interval of penalty factor *C* and kernel function parameter *g*.

As before, the training of each SVM classifier still uses the 10-fold cross-validation method. The classification accuracy of different values of *C* and *g* is visualized in Figure 12, and the maximum values of all metrics are presented in Table 4. Figure 12 shows that parameter *C* has a minimal influence on the accuracy. The accuracy has a trend of increasing and then decreasing with the variation of *g* while *C* keeps steady. The maximum value is 0.930 when *C* = 1.6 and *g* = 31.5.

To design the decision tree, we chose the *C4.5* algorithm and used the postpruning technique to avoid overfitting, and the final results can be seen in Table 4. Regarding the random forests, we needed to study how many trees should be selected to form a forest. First, we chose the MSE (Equation (12)) of the model as a measure to determine the ideal number of trees. Then, a random forest model was trained by the 10-fold cross-validation method, and in one cross-validation, the training set of each tree was 80% of the total. Finally, the average value of ten times was exported as the final evaluation value of the random forest model. From Figure 13, we can see that when the number of trees is greater than 50, and the accuracy and the MSE of the random forest model tends to be stable whether there is data augmentation or not. The MSE of the model is higher than that obtained before data augmentation for a determined random forest model. Meanwhile, the precision accuracy has considerably increased after data augmentation, which means that the DA techniques significantly improved the performance of random forest classification. The maximum value of all metrics is recorded in Table 4.

Figure 14 shows the increment of the four-measure metrics of each algorithm. The results show that not at all algorithms have a significant performance improvement after using the data augmentation. For example, the recall and F1 score of the *LgR* algorithm did not change much. On the other hand, the algorithm with the highest accuracy did not have the most noticeable performance in the other metrics. The accuracy of the GA_SVM is higher than that of FR, but the recall of RF is higher than that of GA_SVM. Meanwhile, despite the high metrics of some algorithms, the positive impact of data augmentation on this is small. For instance, the F1 score of KNN is very close to that of GA_SVM. However, the increment of 0.158 for KNN is lower than that of GA_SVM 0.179. This supports the point that data augmentation technologies have a different impacts on different metrics of algorithms. Therefore, it is worth noticing that different algorithm and metrics should be considered when we use data augmentation to improve the performance of the algorithm.

Figure 15 compares the performance measure for each algorithm with and without data augmentation techniques. From these figures, we can find that the DA positively influences each algorithm, but not every metric of an algorithm is higher than that of the other algorithms. In general, the KNN and GA_SVM have the best classification performance. Logistic regression algorithm has the worst performance. On the other hand, the performance of an algorithm may be significantly improved after use of data augmentation. However, it does not necessarily have the highest classification accuracy. 

## 7. Conclusions and Future Work

In this paper, we propose a novel data augmentation method (AWDBA) to solve the problem of an insufficient training dataset. The results show that the proposed new approach can avoid the influence of anomalous samples in the dataset and generate new data that maintain the features of the raw data. Moreover, we compared the performance of different augmentation techniques. The results indicate that the accuracy of classification using AWDBA is higher than that of other augmentation techniques. We also compare the performance of different algorithms on the augmentation dataset and nonaugmentation dataset. Compared to other algorithms, the KNN and GA_SVM had the better performance. In GA_SVM, we introduced a new tuning strategy based on genetic algorithm to find the optimal penalty factor *C* and the kernel parameter *g*. This method can find the optimal parameters faster than can *GridSearch*. 

It is worth noting that implementing data augmentation helps to reduce the error rate of all algorithms. In addition, the study shows that the KNN algorithm has the best accuracy of classification. However, in others metrics, KNN is not the best. For instance, we can see that the RF algorithm has very high recall. Possible future work related to this study can use more sophisticated techniques, such as a deep neural net, to improve recognition accuracy under more raw data. This will be conducted with consideration to computer performance and a larger training dataset.

## Figures and Tables

**Figure 1 sensors-22-08187-f001:**
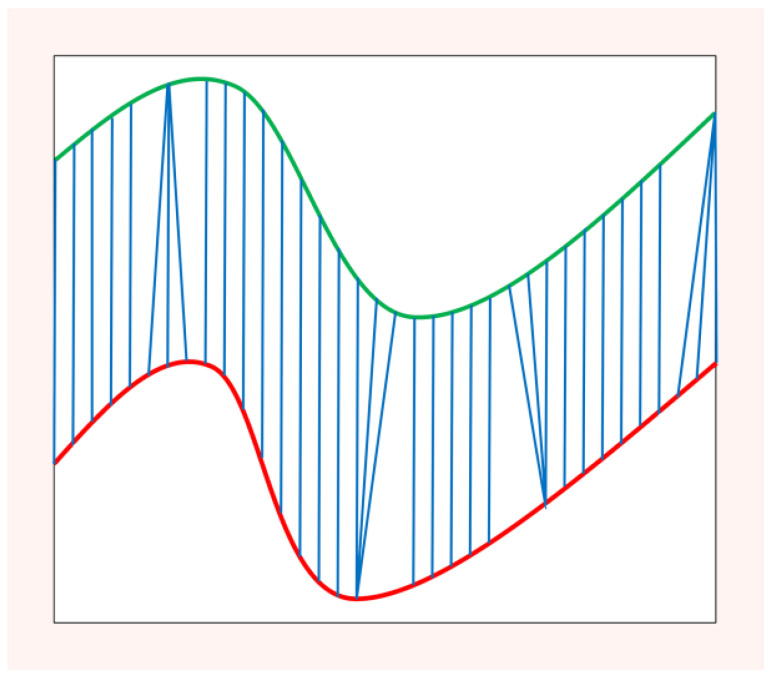
Two sequences aligned with dynamic time warping.

**Figure 2 sensors-22-08187-f002:**
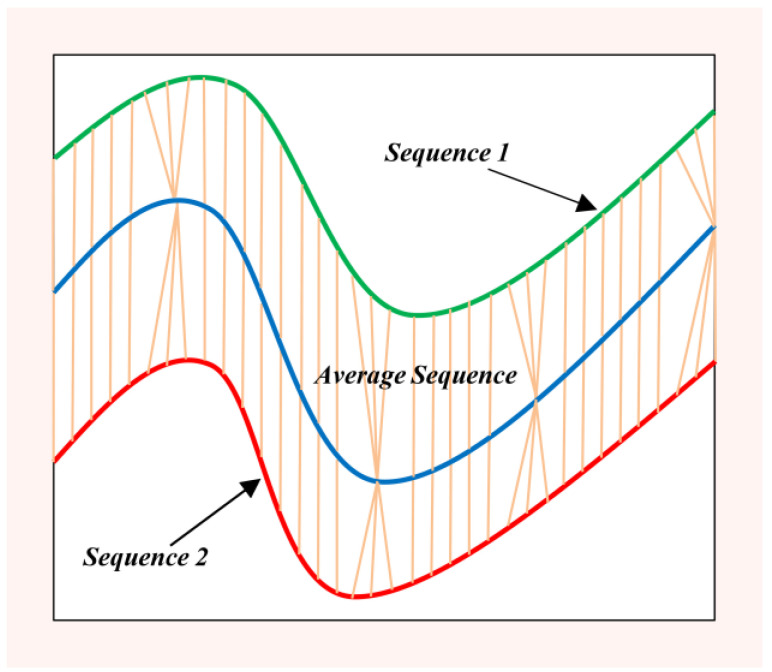
Illustration of the DBA method to acquire the average sequence for two sequences (1 and 2).

**Figure 3 sensors-22-08187-f003:**
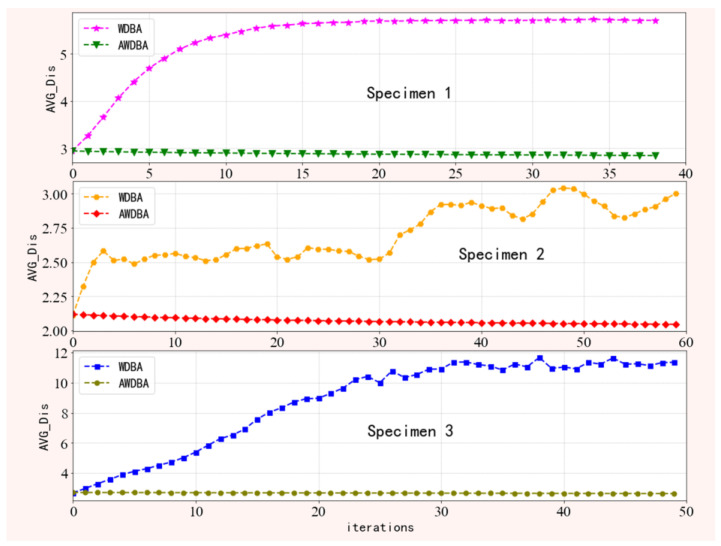
A comparison of the average distance of new data synthesized by WDBA and AWDBA.

**Figure 4 sensors-22-08187-f004:**
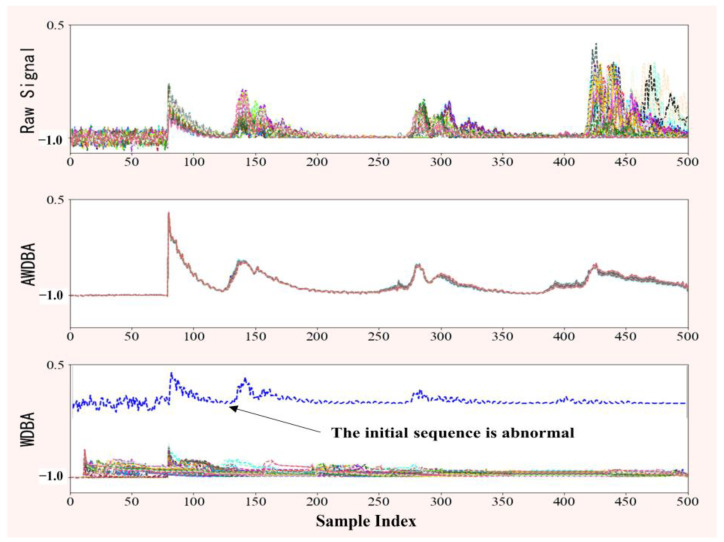
Results of AWDBA and WDBA for data augmentation.

**Figure 5 sensors-22-08187-f005:**
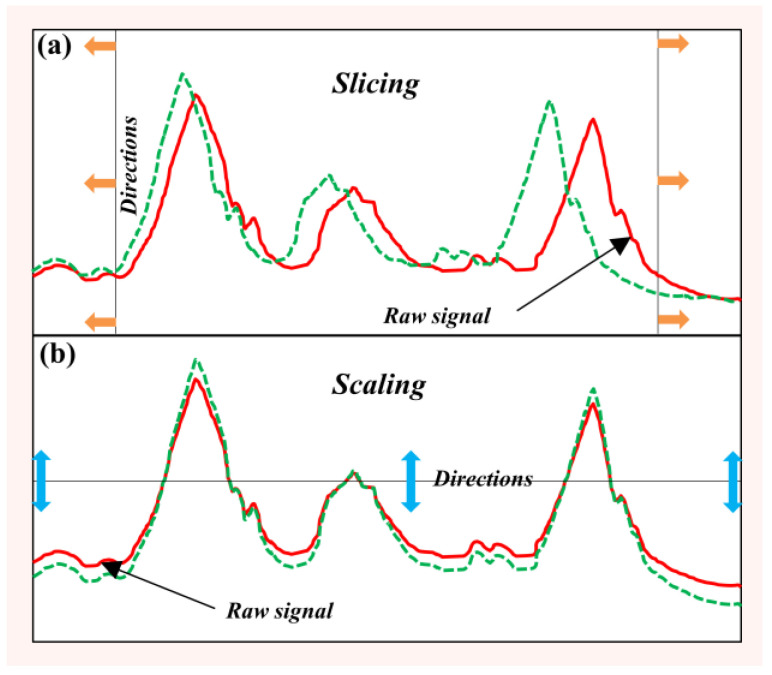
Other time-series data augmentation techniques.

**Figure 6 sensors-22-08187-f006:**
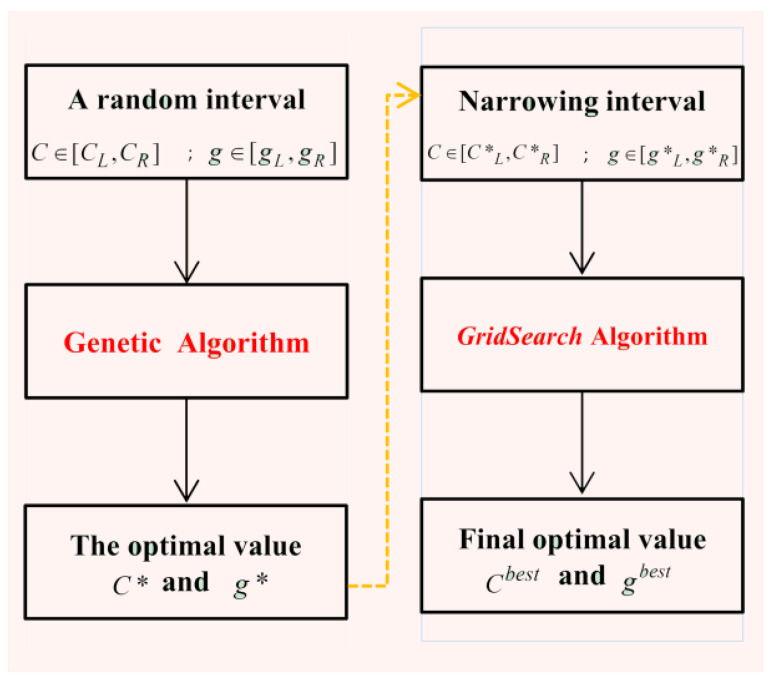
Outline of the proposed *GA_SVM* parameter tuning strategy.

**Figure 7 sensors-22-08187-f007:**
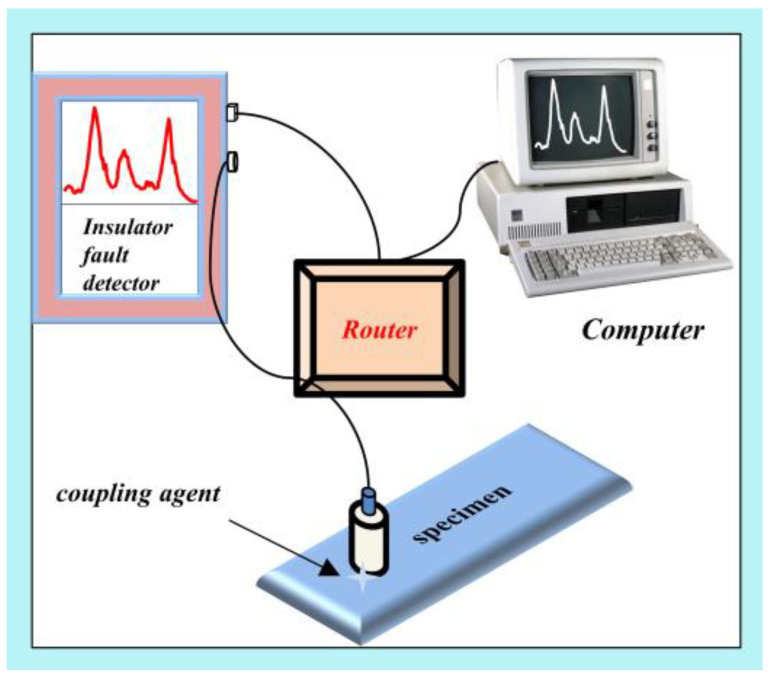
The ultrasonic nondestructive testing of insulator defects.

**Figure 8 sensors-22-08187-f008:**
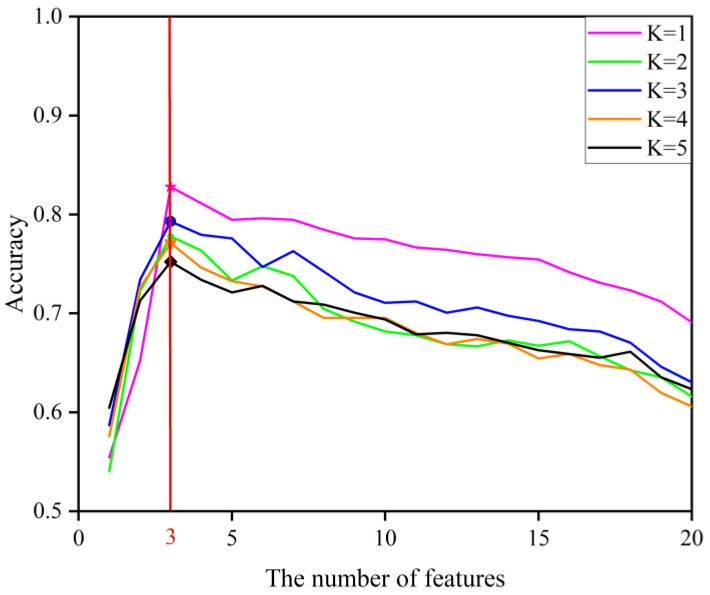
Selected features using sequential floating forward selection algorithm.

**Figure 9 sensors-22-08187-f009:**
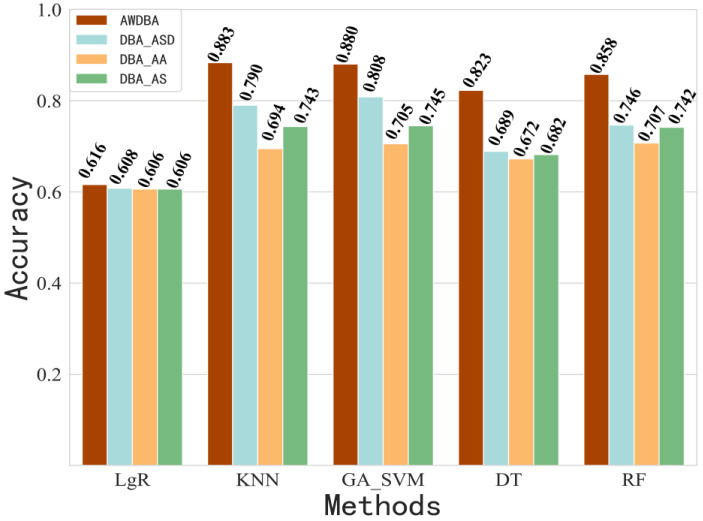
Accuracy of DBA using AA, AS, and ASD weight techniques compared with that of AWDBA.

**Figure 10 sensors-22-08187-f010:**
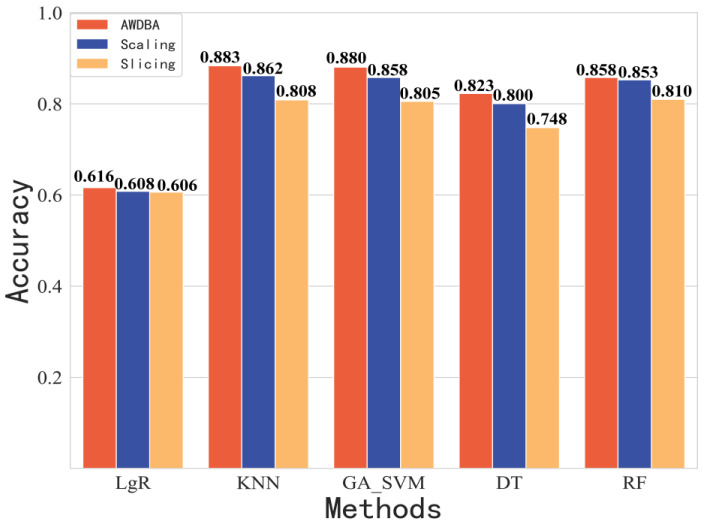
Accuracy of other data augmentation techniques compared with that of AWDBA.

**Figure 11 sensors-22-08187-f011:**
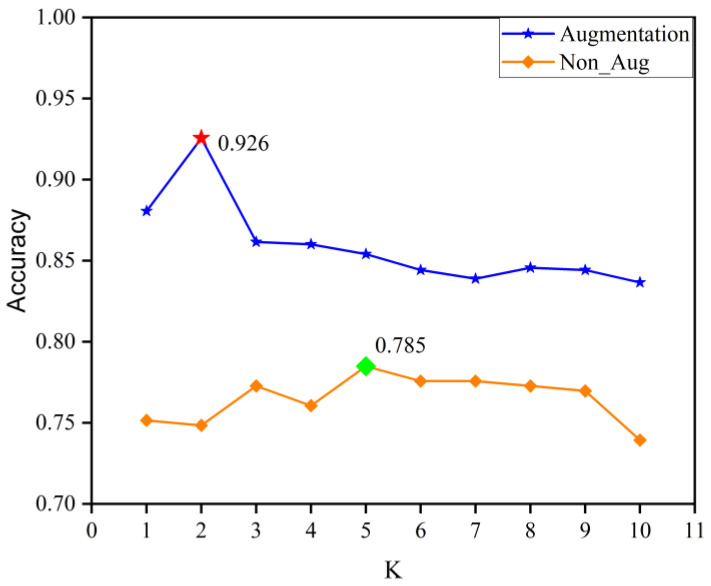
The a accuracy of the KNN algorithm.

**Figure 12 sensors-22-08187-f012:**
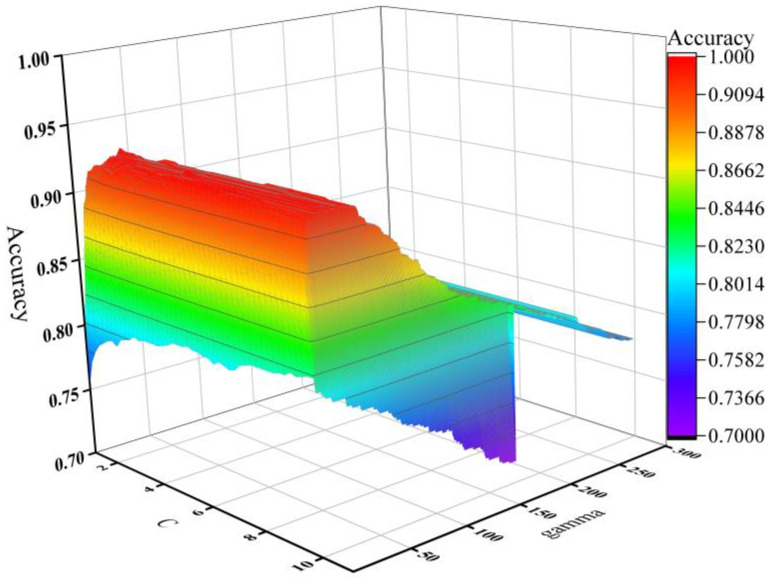
The accuracy of GA_SVM with different *C* and *g* values.

**Figure 13 sensors-22-08187-f013:**
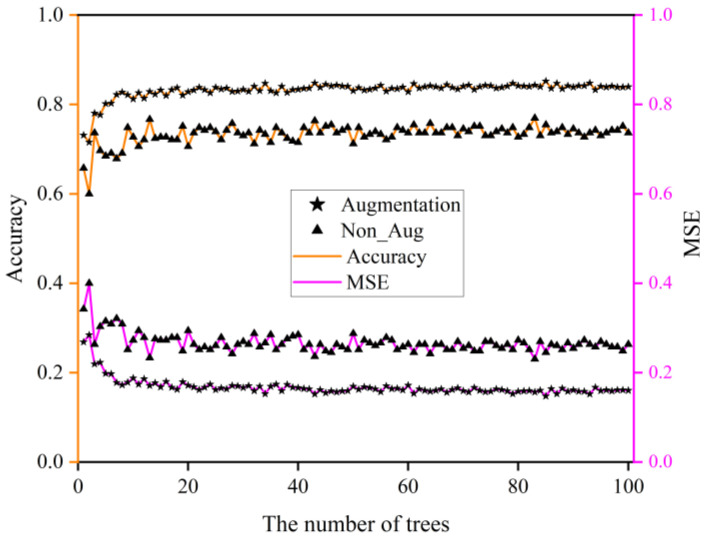
The MSE and accuracy of a random forest model in a different number of trees.

**Figure 14 sensors-22-08187-f014:**
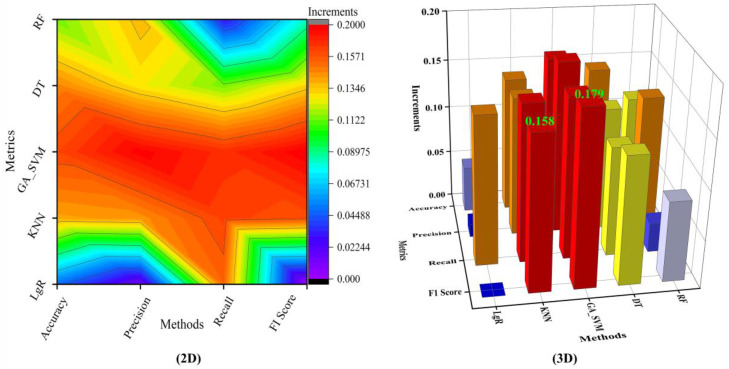
The increments of different algorithm metrics after use of data augmentation.

**Figure 15 sensors-22-08187-f015:**
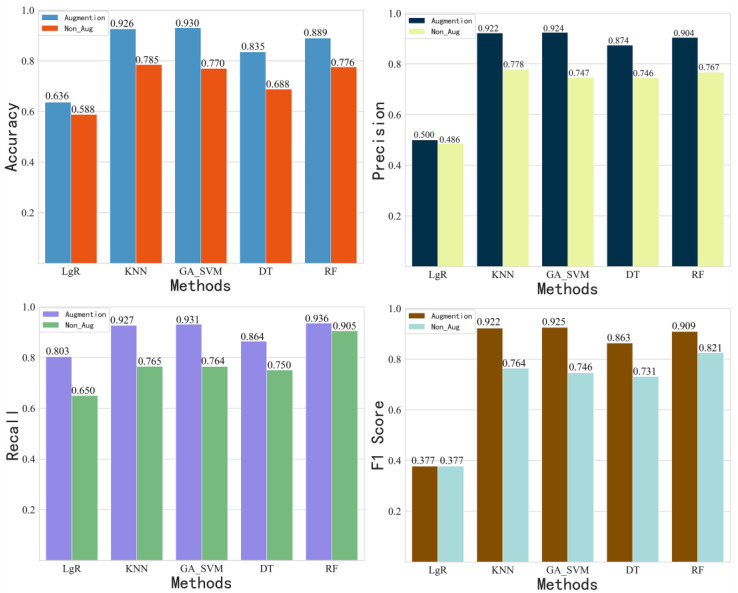
Impact of data augmentation techniques on the performance of the different machine learning algorithms.

**Table 1 sensors-22-08187-t001:** Comparison of augmentation methods for time series.

Methods	Description	Pros and Cons
Jittering, scaling, slicing, and other common methods	These methods rely on the magnitude transformations of the training dataset. Only the values of each element are modified, and the time steps are kept constant.	Simple and easy to operate; however, the characteristics of data distribution are not considered.
Window slicing, time wrapping	These methods are similar to magnitude domain transformations except that the transformation happens on the time axis; the elements of the time series are displaced to different time steps than the original sequence.	Simple and easy to operate, but some important sample points may be lost.
RGANS, RCGANs	Both generators and discriminators are RNN structures.	Difficult to compute in parallel and limited to the medical field.
T-CGANs	Based on one-dimensional CNN, and input is the time stamps.	Adapt to irregular time intervals.
ehrGANs	Use of the one-dimensional convolution GANs to generate medical time series, with semisupervised learning being used.	The classes of new sample and raw sample are as similar as possible.

**Table 2 sensors-22-08187-t002:** Mean value of AVG_Dis for all augmentation data.

	Specimen 1	Specimen 2	Specimen 3
WDBA	5.34	8.88	8.74
AWDBA	2.88	2.07	2.68

**Table 3 sensors-22-08187-t003:** The number of insulator datasets before and after data augmentation.

	Specimen 1	Specimen 2	Specimen 3	3 fold	Total	Label
Defects	39	112	50	603	804	1
Healthy	39	52	40	393	524	0
3 fold	234	492	270	996		
Total	312	656	360		1328	

**Table 4 sensors-22-08187-t004:** The performance measures of all algorithms using data augmentation.

Metrics	LgR	KNN	GA_SVM	DT	RF
Accuracy	0.636	0.926	0.930	0.835	0.885
Precision	0.500	0.922	0.924	0.874	0.905
Recall	0.803	0.927	0.931	0.864	0.936
F1 score	0.377	0.922	0.925	0.863	0.909

**Table 5 sensors-22-08187-t005:** The detailed parameter settings of GA.

Parameters	Settings
The length of gene sequence	10
Population size	150
Maximum iteration	100
Mutation probability	0.8
Breeding probability	0.5
Range of penalty factor C	[1, 10]
Range of kernel function parameter g	[1, 300]

## Data Availability

Not applicable.

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
