# Peer review of "A Novel Data Augmentation Method for Improving the Accuracy of Insulator Health Diagnosis"

_sensors, 2022, doi:10.3390/s22218187_

Round 1

Reviewer 1 Report

In this work, the authors propose a new data augmentation method called AWDBA (Adaptive Weighting Dynamic Time Warping Barycenter Averaging). This method addresses the problem of having insufficient data for training an effective classification and recognition model in the context of intelligent diagnosis of insulators. The authors validate the proposal with two datasets: first, by synthesizing new data from insulator sample datasets, and second, through an original dataset of insulators. A comparison of different machine learning algorithms for insulator health diagnosis on the dataset with and without data augmentation demonstrates that the data augmentation method can significantly improve the accuracy of insulator defect identification.

The main comments to consider regarding this paper are

  • The authors in the paper explain the limitations of classical WDBA. Yet, why do the authors compare against traditional WDTW data augmentation (ASD) and do not include other techniques like Average All (AA) and Average Selected (AS)? Even though the parameters must be chosen manually, the comparison should be included for the completeness of the study.
  • In the paper, as part of their proposal, the authors present a new iteration strategy adopted to avoid the influence of abnormal samples. However, could these abnormal samples be removed through well-known pre-processing methods for outlier removal in time series? See:
    • Chandola, V., Banerjee, A. and Kumar, V., 2009. Anomaly detection: A survey. ACM computing surveys (CSUR)41(3), pp.1-58.
  • The classifiers selected are GA-SVM, KNN, Logistic regression, Decision trees (C4.5), and Random forests. However, the authors never justify why these classifiers are selected. For example, why are multilayer perceptron neural networks excluded?
  • The authors claim that using Genetic Algorithms for hyper-parameter tuning of SVM is a novelty. That is not true. Among many previous works, see for example:
    • Syarif, I., Prugel-Bennett, A. and Wills, G., 2016. SVM parameter optimization using grid search and genetic algorithm to improve classification performance. TELKOMNIKA (Telecommunication Computing Electronics and Control)14(4), pp.1502-1509.
  • Since the authors used 10-fold cross-validation, the bar plots in figures 8 and 13 should include bounds for the distributional limits. Box plots are better than bar plots for this purpose.  

Minor comments:

·       Sections 2.1 and 2.2 have the same title.

·       There is a font size change at the end of page 5.

·       English proofreading is highly recommended. There is sentences long and unnecessarily complex. For example: “Different from other classification methods, decision trees are recursive thinking, it uses binary decision rules to cut off the feature of the training sample…”

Reviewer 2 Report

please find attache our comments. Thank you 

Reviewer 3 Report

In Chapter 6.2 it was stated that the improvement is by using DA techniques. Whether offline DA or online DA techniques were used.

In the conclusions it was stated that deep neural net will be used in further works. This requires the use of more powerful computer hardware. What, according to the authors, will be the required parameters of this equipment and what will be the actual learning time.

Reviewer 4 Report

The authors of the presented work proposed a novel adaptive weighting DBA method and compared its performance with the typical DBA method in improving the accuracy based on different machine learning algorithms for insulator health diagnosis. The results suggest it is a very interesting attempt at data analysis and the method proposed should be very inspiring to readers in related fields.

However, two points must be improved before publication.

 First, the caption or units of the caption of the axis of many figures are lacking, including Fig. 3, 4, 5, 8, 9, 10, and 12. Please carefully check and modify them.

Secondly, the part of “Conclusions and Future work” is too long without highlighting the main findings or output of the study; Instead, it’s more like a description of what the authors did. Please reorganize the core findings, as well as future work, accurately and succinctly.

Round 2

Reviewer 1 Report

The explanation of how Grid Search and Genetic Algorithms are combined is not clear at all. In fact, figure 6 included in the response to the reviewer's comments contradicts the statement that Grid Search is used first and then GA is applied. Please, include a more detailed explanation that matches the figure. 

Reviewer 2 Report

The authors have perfomed significants improvements. However, there are still changes to be made. For example, a paragraph was copied from the paper they add about Feature Selection. please rewrite. 

After this, the paper can be accepted. 
